# Productive Psychoses: Views on Terrorism and Politics in *Homeland*

**Janna Houwen** 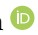

Film and Literary Studies Department, Leiden University Centre for the Arts in Society (LUCAS), Leiden University, Arsenaalstraat 1, 2311 RT Leiden, The Netherlands; j.j.m.houwen@hum.leidenuniv.nl

**Abstract:** In the eight seasons of Showtime's television show *Homeland*, leading character Carrie suffers from a bipolar disorder which repeatedly results in psychotic episodes. During these psychotic breakdowns, her grip on reality is disturbed by delusions. However, her psychotic disposition also leads to abilities and insights that make her a valuable agent in international secret agencies such as the CIA. This essay examines how the *productivity* of Carrie's psychoses can be related to the political, military-industrial order within which she operates as a spy fighting terrorism and other threats to national and international security. What does the fact that a person suffering from psychoses is able to comprehend complex international political processes tell us about these processes and the context in which they occur? To answer this question, I turn to two scholars, both of whom have theorized subjectivity in relation to psychosis: psychoanalyst Jacques Lacan and philosopher Mauricio Lazzarato. The radically different notions of Lacan and Lazzarato lead to different interpretations of *Homeland*. However, although Lazzarato is a critical opponent of Lacanian psychoanalysis, I demonstrate that Lacan's psychoanalytical ideas and Lazzarato's machine theories can to some extent be read as complementary in an analysis of *Homeland*, for what the two distinct theorists have in common is that they both relate subjectivity to sign systems—to the emergence and assignment of meaning, as well as to the suspension and absence thereof. This paper argues that the psychoses of *Homeland*'s lead character produce political meanings because of the condition's specific relation to meaninglessness.

**Keywords:** mental illness; psychosis; madness; terrorism; television series; psychoanalysis; military-industrial complex; semiotics

## 1. Productive Psychoses: Views on Terrorism and Politics in *Homeland*

*'I Was Only Half There'*–Carrie, *Homeland Season 3* ([*Homeland* 2011–2020](#))

Whether they are drugged, tortured, or shot to pieces, characters in the complex fictional world of *Homeland* are regularly dismembered, either psychologically or physically. In the eight seasons of the American television series, which has been running since 2011, we see how secret agents and terrorists constantly subject each other to various forms of surveillance and violence. The lead character, Carrie Mathison, is not only threatened by external enemies. Because the CIA agent (and, later, consultant) suffers from a bipolar disorder, her mental state is far from stable. When the disorder manifests itself, the decisive spy transforms into a raving or catatonic patient, who seems to have lost all control over her statements and actions. During both her manic and depressive episodes, Carrie is plagued by psychoses in which her grip on reality is disturbed by delusions and paranoia. In *Homeland*, the dynamic relationship between Carrie's external enemies and her internal 'demons' serves as more than simply a driver for the plots that unfold in the various episodes of the series. Rather, the lead character's psychoses produce political meanings in *Homeland*.[1] I argue that this is precisely because of the condition's specific relationship to meaninglessness.[2]

Because Carrie's psychoses occur within a context in which tensions and conflicts between Western and non-Western groups reign supreme, *Homeland* initially appears to correspond to a Western discursive tradition that can be traced back to the colonial era. As demonstrated by Ania Loomba in *Colonialism/Postcolonialism* (Loomba 2015), crossing the border between Western and non-Western cultures is often presented as 'dangerous business' in Western tales of colonized areas (Loomba 2015, p. 141). For example, in John Masters' novel *The Deceivers* (Masters 1952), a British officer loses his mind when he infiltrates an Indian gang in the early 19th century, returning to his senses only when he reverts to his Christian identity. In Joseph Conrad's famous *Heart of Darkness* (Conrad 1899), an expedition through the African wilderness drives the hero of the story to madness. The narrator of the autobiographical *Confessions of an English Opium-Eater* (De Quincey [1821] 2008) states during an imperial journey through Asia that he would go mad if he were to live according to Chinese customs for too long (De Quincey [1821] 2008, pp. 72–73). In summary, those who immerse themselves excessively in non-Western countries and cultures run the risk of going mad, due to contact with the wild and primitive. Stated even more strongly, 'Africa, India, China and other alien lands induce madness, they *are* madness itself' (Loomba 2015, p. 141).

As a secret agent, Carrie Mathison performs tasks that, in addition to the physical danger to which she is exposed, could certainly be regarded as 'dangerous business' when viewed from within the aforementioned Western tradition. In order to infiltrate non-Western cultures and recruit 'assets,' Mathison is uniquely aware of how she must invisibly assimilate into other cultures. Her psychoses could thus also be interpreted as logical consequences of her professional tasks: undercover missions in primitive, underdeveloped and barbaric countries cause madness. In this interpretation Carrie's psychoses take on a political meaning. Her madness means that non-Western cultures are presented in *Homeland* as disturbed, undeveloped, and dangerous. Such a conclusion chimes in with the deluge of criticism that the series has received. Critics have referred to *Homeland* as 'TV's most Islamophobic show' (Al-Arian 2012, n.p.), accusing it of confirming American 'post-9/11 insecurities' and all the stereotypes that go with them (Castonguay 2015, pp. 140–43). They have also associated the television series with 'US propaganda that returns to an age-old Orientalist tactic that pathologizes Islam and constructs it as something that needs "curing" or Western "enlightenment"' (Razack and Maira in Bevan 2015, pp. 148–49).

Although orientalist and racist stereotypes can undoubtedly be identified in *Homeland*, a reading of Carrie's psychoses as consequence of cross-border contact with non-Western cultures is problematic, for two reasons. First of all, the border between Western and non-Western is blurred in the television series, as the binary oppositions that have been used throughout history to construct the difference between the two (e.g., healthy/sick, civilized/primitive, rational/emotional, adult/child, White/Black, Christian/Muslim, good/bad, victim/perpetrator) are both confirmed and deconstructed in *Homeland*. For example, white American citizens turn out to be Muslim terrorists, violent Taliban and al-Qaida leaders are presented as victims of the USA, CIA agents desert or take irrational decisions, and jihadists prevent violence. Moreover, all the depicted groups deploy underhanded strategies that force or seduce characters to switch identities, and the military-industrial arena in which political conflicts are fought out turns out to comprise so many parties that a simple division between 'the West and the rest' is in no way applicable. As Jason Mittell has noted, the instability of meanings in *Homeland* is in part enabled by the serial form, which allows leeway for a continuous process of revision and re-contextualization (Mittell 2015, p. 346). Within the 'semiotic trap' that the series forms (Letort 2017, p. 164), it is impossible to attribute Carrie's psychotic delusions unequivocally to the madness of the cultural others that she meets.

Second, this causal interpretation of Carrie's psychoses is untenable because throughout the seasons of *Homeland*, she usually decides on her own to discontinue her medications and, subsequently, to make voluntary descents into psychosis. For Carrie, the transition to a manic-psychotic state is an attractive process. As she explains to her partner Jonas in

Season 5, the transition from 'making sense' to incomprehensible 'incoherence' includes a period in which she sees everything clearly and is able to perform at her best. Within her position as a secret agent and consultant, her pathology is thus productive to some extent. In addition to terrible anxieties and confusion, it provides Carrie with insights that she does not have in her medicated state.[3]

The question that arises at this point is as follows: how can the *productivity* of Carrie's psychoses be related to the political, military-industrial order within which she operates? In other words, what does the fact that a person suffering from psychoses is able to comprehend complex international processes tell us about these processes and the context in which they occur? To answer this question, I turn to two scholars, both of whom have theorized subjectivity in relation to psychoses: psychoanalyst Jacques Lacan and philosopher Mauricio Lazzarato. The radically different notions of Lacan and Lazzarato lead to different interpretations of *Homeland*, which nevertheless both contribute to answering my question in specific ways. Although Lazzarato is a critical opponent of Lacanian psychoanalysis, I will demonstrate that Lacan's psychoanalytical ideas and Lazzarato's machine theories can to some extent be read as complementary in an analysis of *Homeland*, for what the two distinct theorists have in common is that their approaches are *semiotic*. Both Lacan and Lazzarato relate subjectivity to sign systems—to the emergence and assignment of meaning as well as the suspension and absence thereof. In addition to allowing a critical comparison of the two scholars, this semiotic grounding of their approaches is especially suitable for an analysis of *Homeland*, in which both the lead character and the fictional world itself are characterized (and ravaged) by dissolving meanings.

## 2. Psychoanalysis: A Lacanian Approach to *Homeland*

Lacan's explanation of the emergence of psychoses is associated with his assumption that the subject is an effect of language.[4] Within chains of signifiers, the subject is produced by 'shifters' (e.g., 'I' and 'you'), which represent the subject but that do not present it or render it present, as the speaking subject can never coincide with the grammatical subject. In Lacan's theory, subjectivity is not a permanent or constant state, but something that constantly recurs through a certain composition of signifiers: 'the subject is always a fading thing that runs under the chain of signifiers' (Lacan 1970, p. 194). Despite the volatility of this process of subjectivization and the divisions inherently resulting from it, entry into the symbolic order is a necessary condition for acquiring an identity and individual desire. For those suffering from psychoses, the passage to the symbolic order has not occurred.

In most cases, this entry into the symbolic order is a gradual process in the development of children, in which the child becomes separated from the mother through the intervention of the father. As expressed by Antoine Mooij, the father intervenes 'by setting the law, or rather, by expressing the law' (Mooij 1987, p. 135, own translation). Because the emergence of the separation does not necessarily involve a father who is physically present, but can also be a reference to the father in the narrative of the mother, Lacan does not speak of the father, but of the 'Name-of-the-Father' (Mooij 1987, p. 137). This is a symbolic function that expresses both a special law (the prohibition of incest) and the idea of rules or laws as such. The acceptance of the Name-of-the-Father is an acceptance of the prevailing rules—an acceptance of linguistic–social rules and differences that together serve as the foundation for the symbolic order. When children relinquish their imaginary identification with the desire of the mother and exchange it for an acceptance of the Name-of-the-Father, the symbolic takes on an ordered structure, so that the subject can be articulated in a consistent manner (Vanheule and Geldhof 2014, p. 77). Through the Name-of-the-Father, 'the subject can consider himself and others from within a framework of rules and standards that must be met. It is a signifier that makes desire comprehensible and promotes a sense of stability in the perception of social relationships' (Vanheule and Geldhof 2014, p. 77; own translation).

According to Lacan, people with psychoses have for some reason rejected the Name-of-the-Father, which leaves 'a hole in the symbolic' (Lacan [1955] 1993, p. 156). As a



consequence, an individual with psychoses cannot assume a subject position in relation to others, because the social-cultural rules that enable this have not been accepted. This makes it hard for people who suffer from psychosis to understand and interpret the intentions of others. Moreover, they lack the experience of a having a stable individual identity. Without a framework of socio-cultural rules, questions like 'Who am I?' and 'What do you want from me?' cannot be answered in a conventional manner (Vanheule 2011, p. 68). Therefore, intimate relationships and confrontations with others are often confusing and intimidating for people with psychoses. They always remain—at least to some extent—'outsiders' who do not feel connected to a group.

### 2.1. Carrie's Rejections

The first four seasons of *Homeland* increasingly sustain a Lacanian reading of Carrie's psychoses, as the information about Carrie's parents, which is revealed gradually, makes a rejection of the Name-of-the-Father plausible.[5] Carrie's father also suffers from a bipolar disorder and, in Seasons 1 and 2, he supports her with advice and assistance when she sinks into and claws her way out of a manic-depressive period with psychotic characteristics. Carrie nevertheless does not accept his recommendations as a father and 'experiential expert.' She rejects his authority time and again. The rejection of paternal authority does not necessarily equate to the rejection of the paternal signifier. The latter does seem probable, however, when Carrie learns in Season 4 that, years ago, her mother had left the family because she was pregnant by her lover, after a long series of affairs with various men. Carrie's acceptance of the Name-of-the-Father was possibly obstructed by the position of her mother in relation to her father.

When we take later notions of Lacan on the Name-of-the-Father into account, Carrie's rejection of the paternal signifier in *Homeland* can also be recognized outside of her family relationships. In the 1960s, Lacan stated that the Name-of-the-Father is not so much a unique signifier with inherent characteristics as it is a uniquely used signifier in which people believe (Vanheule and Geldhof 2014, p. 80). From that time onwards, Lacan refers to *a* Name-of-the-Father, arguing the possible existence of multiple paternal signifiers ('Names-of-Fathers'), which are elevated to structuring Symbolic elements through an act of faith. The rules and explanations of a person with authority are accepted as law through an act of faith. In individuals with a psychotic structure, however, this act of faith is missing: 'believing is not an issue in psychosis' (Lacan in Vanheule 2011, p. 136).

In the seven seasons of *Homeland*, Carrie's rejection of possible paternal signifiers (Names-of-Fathers) is continually repeated. In the stories of *Homeland*, the male and female characters who serve as paternal signifiers for large groups of people by virtue of the fact that they stand as figures of authority for a particular order or law (e.g., the various successive CIA bosses, judges, CIA station chiefs and ambassadors) are not accepted as such by Carrie. She keeps evading the rules, laws, and assignments that originate from these figures. For example, as a CIA agent, she independently carries out illegal surveillance operations, lies to the Senate under oath, and blackmails CIA boss Lockheart to advance her own career. The outsider position that Carrie repeatedly assumes through her disobedience is especially confirmed when, in Season 5, she even turns away from mentor Saul Berenson, in whose ideas she often invested considerable confidence. In addition, she turns against the female president of the USA, whom she appears to accept as a paternal signifier in Season 6, but eventually fights as an enemy through undercover missions in Season 7.

Carrie's lack of blind faith in others—which, in a Lacanian reading, could be interpreted as a rejection of Names-of-Fathers—is in one sense a disadvantage in her work as a spy and security consultant. In undercover missions, her fate is in the hands of her colleagues, whom she must trust in serious predicaments. By ignoring recommendations or agreements, she sometimes puts herself or others in danger. Moreover, as a person with psychoses, she has difficulty estimating the intentions of others, even though her task specifically involves predicting violent attacks. Despite her mistrusting attitude, she is often taken aback by the decisions of others, leaving her devasted and emotional.

At the same time, however, her position as an outsider works to her advantage. When we read Carrie's pathology with Lacan, we can say that her instable subject position prevents her from interpreting social relationships and the intentions of others in a conventional manner through the symbolic order. Consequently, she takes scenarios into account that no one else could ever anticipate. Carrie is, for example, the only one who manages to see deserted marine sergeant Nick Brody outside of the customary framework of a White, male American war hero, and her suspicions of other seemingly 'good Americans' (like Aileen Morgan and her husband Raquim Faisel in Season 2, and the ambassador's husband, Dennis Boyd, in Season 4) often prove to be reasonable.

### 2.2. Imaginary Crutches

Carrie's strength as a spy does not depend solely on the unconventional view afforded by her instable identity. It also depends on the way in which she continuously manages to stabilize that identity. Although the psychotic structure of individuals after the rejection of the Name-of-the-Father cannot be 'healed,' people with psychoses can achieve a certain level of stability by adopting various means of compensating for the missing Name-of-the-Father. In his third seminar, Lacan discusses the principle of conformist imaginary identification, in which the psychotic individual blindly copies the behaviour and lifestyle of particular people or groups. Through a form of identification that Lacan refers to as 'imaginary crutches' (Lacan [1955] 1993, p. 205), psychotic individuals provide themselves with models of behaviours for particular social roles. These models are copied without any questions on identity (is this me?) or morality (is this good or bad?). As emphasized by Vanheule, conformist imaginary identifications are 'remarkable for their lack of affectivity, their uncritical nature and their lack of subjective implication' (Vanheule 2011, p. 73).

In a Lacanian reading of *Homeland*, Carrie's identification with particular roles can be recognized as the cool rigidity with which psychotic people copy behaviour when they take on 'imaginary crutches.' When Carrie interrogates suspects and recruits 'assets,' she has no qualms about becoming the 'good cop' who 'breaks' a suspect. She can turn into an understanding friend in order to gather information from the wife of a Hezbollah leader, or play the part of a comforting lover, thereby gaining the trust of a young Pakistani student who could be useful to the CIA. Carrie's instrumental imitation of empathic behaviour disappears once she is promoted to the position of station chief in Islamabad. There, Carrie follows a new set of unwritten rules associated with her new management position: she keeps a cool eye on broader interests and decides on airborne attacks in a detached, rational manner. Although Carrie receives loathing and an infamous nickname ('the drone queen') by coolly adopting different roles, it cannot be denied that she also owes her success as a secret agent to the various 'imaginary crutches' on which she leans without hesitation.

### 2.3. Decompensation

After long periods of compensation by conformist identification, people suffering from psychoses can nevertheless suddenly decompensate, and subsequently descent into a psychosis (Lacan [1955] 1993, p. 205). During a manifest psychosis, the absence of a stable subject position is expressed in an experience of the 'inmixing of subjects' (Lacan [1955] 1993, p. 193)—a merging of the boundaries between self and others—in which the tipping point into delusion can be identified at the point at which the person with psychosis has the notion that the Other is taking the initiative: '*The Other wants this*' (Lacan [1955] 1993, p. 193).

One of the most remarkable aspects of Lacan's approach to psychoses is that, in contrast to many other psychoanalysts and psychiatrists, he asserts that psychotic delusions and hallucinations are not due to neurological defects or warped sensory perceptions of reality, but that they emerge from interruptions in processes of meaning production. For example, according to Lacan, hallucinations are interruptions of total foreignness in the Real, which are caused by the absence of the Name-of-the-Father in the symbolic order. Through this hole in the symbolic, the person experiencing a psychosis is sometimes

confronted with matters that cannot be symbolized through copied norms and ideas. This could lead to the appearance of an incomprehensible, unchained signifier in the Real.

In addition to hallucinations, another symptom that emerges from disturbed processes of meaning production is the overwhelming delusional experience that there *is* meaning, or that certain objects *have* meaning, although psychotic people do not know exactly which meaning that is. Lacan argues, however, that psychosis is not solely produced on the level of meaning, but 'also that it essentially stems from something that is situated at the level of the subject's relations with the signifier' (Lacan [1955] 1993, p. 199). Through the hole in the symbolic order, the metonymic references between chains of signifiers that would normally be stabilized and structured by the Name-of-the-Father are disturbed. This has implications for the relation between the relatively autonomous domains of signifiers and the signified. When we use language, according to Lacan, so called 'quilting points' arise—intersections at which signifiers are briefly linked to the signified. In manifest psychoses, however, these intersections are absent, with the consequence that 'the signifier and signified present themselves in completely divided form' (Lacan [1955] 1993, p. 268). For people with psychoses, language is an enigmatic autonomous fact, something that 'speaks all by itself, out loud, in its noise and furor, as well as in its neutrality' (Lacan [1955] 1993, p. 250). According to Lacan, the nucleus of psychosis resides in the relationship between the psychotic subject and the signifier in its most formal dimension, and the affective reactions of the psychotic to that specific relationship to the signifier (Lacan [1955] 1993, p. 250). The psychotic experience that everything is highly meaningful is caused by the fact that the signified continues to slip under the chain of signifiers. As explained by Vanheule, a person going through a psychotic episode thus has an intuitive sense that many things are happening at the level of the signified, but there are no words—no signifiers—for it (Vanheule 2011, p. 104).

According to Lacan, the emergence of a psychosis plays out between two poles: on the one hand, day-to-day language usage, and, on the other hand, a revealing side of the signifier: the signifier as a 'word of revelation.' Without 'quilting points,' ordinary language usage is insufficient to express the experiences of an 'abundance of modes of being' and the 'intermixing of subjects' that accompany a psychosis. A person experiencing a psychosis 'lingers over a mere shell, an envelope, a shadow, the form of speech' (Lacan [1955] 1993, p. 254). At the same time, however, the person experiences a side of the signifier that is characterized by a certain completeness:

> . . . a side of the signifier that is given to us for its particular density–not for its meaning, but for its meaningfulness. The signified is empty, the signifier is retained for its purely formal properties, which are used to form series. This is the language of the birds from the sky, the discourse of the young girls [ . . . ]. (Lacan [1955] 1993, p. 255)

This specific experience of a meaningless yet meaningful signifier disappears in later stages of a psychosis. The signifier ultimately falls into complete silence in a person who is experiencing a psychosis (Lacan [1959] 2006, p. 468), although occasional emergences of 'non-sensical signifieds' can overwhelm the psychotic person as an aftershock of the process of meaning production, which has come to a standstill. If the metonymic chains of signifiers are brought to a standstill, the complete meaninglessness of language can be expressed in screams and growls, which reveal the 'absolutely a-signifying vocal function' (Lacan [1955] 1993, p. 140) of language.

### 2.4. 'It's Bright Purple'

Carrie's symptoms during her breakdowns in *Homeland* largely correspond to Lacan's discussions of the manifest psychosis. Her hallucinations, in which she sees the deceased Brody and Ayaan, could be interpreted as manifestations of the fact that Carrie is unable to symbolize the loss of these two individuals. She cannot relate to it as a subject—partly due to existential questions that she has in response to her own share in the death of these two men. Her anxious, incomprehensible monologues and the sheets of paper that she

fills with scribbles at the beginning of a manic psychosis could be regarded as frantic attempts to communicate experiences and ideas for which there are no words. In her manic-psychotic state, Carrie feels that there is a lot of meaning: she announces that something very important should be noticed. Yet the meanings slip away, and Carrie ultimately falls silent.

*Homeland*'s representation of Carrie's psychotic episodes seems to illustrate Lacan's idea that psychosis emerges between two poles, in which language usage is separated from meaning, but 'empty' signifiers nevertheless remain highly meaningful for psychotics, despite the absence of signifieds. Carrie emphatically refers to the abstract collages, coloured notes, sheets of paper covered in writing, and photographs that she gathers around herself when sliding into a psychosis as 'very meaningful,' but they cannot be related to signifieds. Warnings like 'It's bright purple!' concerning a particular security situation do convey importance, but no meaning. After a period of rest, silence, medication, or a little help from her mentor Saul, Carrie is nevertheless able to arrive at insights through her complicated assemblies. She suddenly sees connections or processes in international terrorist networks, and her chaotic collages turn out to make sense in some way. For the time being, however, it remains unclear exactly why the period (or, as Carrie refers to it, her 'window') of moving up and down between meaningfulness and meaninglessness helps to generate insight into terrorist organizations.

In a Lacanian reading of *Homeland*, Carrie's success as a secret agent can be explained largely in terms of her instable subject position. When her psychoses are seen through the lens of Lacan's theory, we can see how her 'madness' provides her with an unconventional view and leads to useful compensation techniques. However, the fact that Carrie excels because she never takes others for who they appear to be, and is able to shift between roles in a chameleon-like manner, does not just paint a specific picture of psychosis as a pathology. It also demonstrates how the political, military-industrial order of security services depends on shifting, ambiguous, and instable identities. It is even possible to discern a psychotic structure in the world of security services that is depicted in *Homeland*. As remarked by Anat Zanger, there seems to be a 'rupture in the patriarchal order' (Zanger 2015, p. 739) in *Homeland*, as Names-of-Fathers appear to function only weakly, if at all. The world of *Homeland* is a world devoid of stable symbolic fathers, and it is thus also devoid of a stable, generally accepted law. The CIA is represented as a hierarchical, patriarchal organization, yet it is also shown to regularly operate outside of legal, socio-cultural, and moral laws. The American nation itself appears to suffer from a prevailing lawlessness in *Homeland*, with presidents and vice-presidents operating outside of the rule of law. The terrorist groups in *Homeland* do not suffer from weak Fathers, but their ideological patriarchal structures are rendered instable by the moral doubts of members. Within a Lacanian analysis of *Homeland*'s diegesis, Carrie's psychosis constitutes a *mise en abyme*. Her pathology provides a small-scale reflection of the disorder suffered by contemporary post-9/11 societies and related military-political groups, which makes Carrie very much compatible with her context. However, this Lacanian analysis does not yet fully explain why Carrie's psychotic episodes lead to such important insights on cases that she is working on. In order to relate the *productivity* of Carrie's manifest psychoses—and particularly her 'window' between meaningfulness and meaninglessness—to the international military-industrial order within which she operates, it is helpful to consider *Homeland* from a completely different theoretical perspective: that of Maurizio Lazzarato.

## 3. Machine-Analysis: A Reading of *Homeland* with Lazzarato

'The danger is coming!', writes Maurizio Lazzarato, after identifying the existence of a conscious and unconscious return to Freud and Lacan in his article 'Some "Misunderstandings" on Desire' (Lazzarato 2017). According to Lazzarato, Lacanian psychoanalysts are currently stealing the show in the Italian media 'by lamenting the society without fathers, the dissolution of "good old patriarchy" caused by the vaporizing of the father' (Lazzarato 2017, p. 50). Lazzarato sees this in line with the French extreme right-wing call for restoring

the honour of the nuclear family, which he defines as an alarming return to a prototype of the family that originated from the neurotic Bourgeoisie of the 19th century. When *Homeland* is read with Lacan as a (negatively tinted) representation of a world without fathers, the show could be said to chime in with the dangerous contemporary tendency that Lazzarato identifies.

According to Lazzarato, however, the danger of a return to psychoanalytic notions does not lie solely in the revival of restrictive and repressive norms and values (which Lacan, does not propagate either; he analyses them as a structure). Lazzarato argues that Lacanian, psychoanalytic analyses are particularly inadequate in political terms. The Italian philosopher asserts that important forms of political-economic repression and subjection are left out of consideration when the emergence of the subject is studied in a Lacanian manner. When the subject is interpreted exclusively as an effect of language, and when language is the source of the subject, political scholars who react or build upon the philosophies of Lacan in a critical manner (e.g., Judith Butler, Slavoj Žižek, and Jacques Rancière) remain trapped in a logocentric world (Lazzarato 2014, pp. 58–60). According to Lazzarato, such logocentrism does not correspond to contemporary political-economic reality, because 'with capitalism we have for some time entered a "machine-centric" world that configures the functions of language in a different way' (Lazzarato 2014, p. 60).

In order to analyse and criticize the contemporary machine-centric world, Lazzarato posits various stages of subjectivation in his book *Signs and Machines* (Lazzarato 2014). In these stages, a developing 'self' continuously relates to signs in differing ways. The final stage, in which the experience of a verbal self emerges through and in language usage, could be compared to the individual's entry into the symbolic order in Lacan's theory. Prior to this entry into language, however, Lazzarato proposes forms of 'preverbal senses of self,' or non-verbal parts of subjectivity that also continue to be present after the emergence of a verbal subject. Because self-positioning is phatic and affective before becoming linguistic and cognitive, the development of the subject is both discursive and 'situated at the focal point of (existential) non-discursivity at the core of subjectivity' (Lazzarato 2014, p. 102).

In addition to claiming that that subjectivity exists prior to and without language, Lazzarato argues that sign systems are operational before subjectivity becomes verbal and individual. Before children acquire a verbal language, they construct forms of observation and communication through highly rich, differentiated non-verbal semiotization. An emerging self-awareness is then accompanied by what Lazzarato refers to as 'asignifying semiotics': intensities, temporal forms, rhythms and movements, that are recognized and affectively understood by small children. In the absence of language, babies organize and select what is going on around them, even though they are not yet aware of any distinction between subject and object. Communication occurs through contagion: 'ways of feeling' are passed along without words. Lazzarato writes that this pre-individual phase and its associated 'asignifying modes of semiotization' can form a reservoir of creativity and learning capacity in later stages of development (Lazzarato 2014, pp. 103–4). Even more important to Lazzarato, however, are the political implications of pre-individual subjectivity, 'since this same pre-individual subjectivity is brought to bear by capitalist machinic enslavements' (Lazzarato 2014, p. 102).

### 3.1. Forms of Subjection

Following the theories of Deleuze and Guattari on machines, Lazzarato does not interpret the machinic in purely technological terms. Machines do not depend on *techne* alone; they can also consist of aesthetic, economic, or social components. In his work, Lazzarato analyses the capitalist system as a machine that is composed of economic, social, and technological elements. These elements form an assembly of coherent tools that subject the user of the machine.

The aspect of subjection—which, in Lazzarato's theory, is inextricably bound to the machinic—consists of two parts. Lazzarato draws a distinction between *subjection* and *enslavement* by the machine. *Subjection* means that the machine creates social roles and

functions, endowing us with subjectivity and allowing for individualization through categories, like nationality, sex, or occupation (Lazzarato 2006, p. 1). Through meaning-producing elements ('semiologies of signification') which create differentiated roles, we are made into subjects according to the demands of power, and each individual receives a unique, unchangeable identity within and through the machine (Lazzarato 2006, p. 1).

More important to an analysis of *Homeland* is the process that accompanies this production of the individualized subject: the process of de-subjectification to which Lazzarato refers as *machinic enslavement*. Machinic enslavement dismantles the individualized subject—the set of pre-individual components of subjectivity in operation (Lazzarato 2006, p. 3). We are transformed into slaves by a machine when we act as a 'cog in the wheels' or as 'one of the constituent parts enabling the machine to function' (Lazzarato 2006, p. 1). The ontological boundary between subject and object, as erected in processes of subjection, is rendered porous through machinic enslavement: 'machinic enslavement [ . . . ] considers individuals and machines as open multiplicities. The individual and the machine are sets of elements, affects, organs, flux and functions, all of which operate on the same level and which cannot be articulated as binary oppositions' (Lazzarato 2006, p. 4).

This dissolution of boundaries and binary oppositions does not occur through language, but through what Lazzarato refers to as 'a-signifying semiotics.' A-signifiying signs do not denote; as so-called 'power signs' they work directly on material components (Lazzarato 2014, p. 85). For example, Lazzarato mentions computer languages that drive technological machines, share prices that influence production processes, and codes on parking tickets that open the gates to the car park (Lazzarato 2014, pp. 85–86). In the realm of asignifying semiotics, a stream of signs provides an input and output of data, albeit without producing meanings: 'bypassing signification' (Lazzarato 2014, p. 85).

Following Guattari, Lazzarato refers to the operation of asignifying semiotics as 'diagrammatic' (Lazzarato 2014, p. 86). Guattari defines the diagram as a sign that has operational instead of representational functions. According to Lazzarato, diagrams—including equations, tables, and graphs, as well as apparatuses and buildings (e.g., Foucault's panopticon)—do not represent, but re-*produce* the functioning of a machinic system. Through asignifying semiotics, the machine engages in interaction with comparable 'agents of partial discursivity,' like atomic, biologic, and chemical strata (Lazzarato 2014, p. 88). Moreover, the machine operates through asignifying semiotics in the same way as the pre-verbal and pre-individual worlds of human subjectivity, in which intensities, temporal forms, rhythms and movements are not grasped by language, but are part of an asignifying semiotic system.

*3.2. Loss and Enrichment*

The parallel between the asignifying semiotics of the machine and the asignifying semiotics that prevail at the level of pre-individual subjectivity is important when analysing *Homeland* from the perspective of Lazzarato's theory. Unlike Lacan, Lazzarato has not written any detailed studies on psychosis, yet a brief remark from the Italian philosopher plays a key role in identifying Carrie's pathology in relation to the fictional world of *Homeland*. According to Lazzarato, we mainly have access to asignifying semiotics during our childhood, when the first forms of non-verbal semiotization emerge. However, later on in life, such access is also possible though psychosis (Lazzarato 2014, p. 104).[6] In *Homeland*, Carrie's insights during her psychoses could thus be attributed to the fact that, during her breakdowns, she gains access to a form of semiotization that is characteristic of both a pre-individual level of subjectivity and the subjecting, enslaving machine.

Lazzarato does not explain why psychosis exactly provides access to asignifying semiotics. Interestingly, Lacan's afore discussed outline of psychotic symptoms can be of help here. Certain characteristics of the pre-individual level and the associated asignifying semiotics correspond at least to some extent with the experiences of the psychotic individual, as discussed in Lacan's third seminar. The experience of an 'inmixing of subjects' that Lacan relates to the beginning of a psychotic episode resembles the absence of delineated subjects

and objects at the pre-individual level, which are called upon in the case of machinic subjection. Although Lazzarato does not speak of signifiers without the signified but of signs without meanings, his remark that the asignifying signs of the machine are in interaction with atomic, biological, and chemical strata recalls Lacan's proposition that, in a preliminary stage of a psychosis, a person experiencing a psychosis has access to a full signifier that is used separately from the signified in order to form series—series like the language of birds in the air. Despite the fact that Lazzarato radically contradicts Lacan's notions on the Name-of-the-Father and generally accuses psychoanalysis of an excessive focus on language and meaning (or signifying semiotics), there appears to be a general correspondence between the two theoreticians when it comes to psychosis, as both scholars relate the pathology to meaningless signs.

Nevertheless, an interpretation of *Homeland* based on Lazzarato's theory instead of Lacan's ultimately leads to a different view of Carrie's psychoses, as well as of the world in which she operates. Whereas Lacan reads psychosis as a pathology in which subjectivity is instable and the experience of a self disappears completely during the course of a manifest psychosis—which Lacan even describes as the death of the subject (Lacan [1959] 2006, p. 473)—Lazzarato interprets a psychosis as the manifestation of a different level of subjectivity. Instead of the loss of subjectivity, Lazzarato relates psychosis to the pre-individual level or to an emerging self outside of consciousness and language.

On the one hand, *Homeland* represents Carrie's breakdowns as a process of loss in when she is portrayed as a catatonic, hospitalized medical object (Season 3) or as a patient wandering around in hallucinations, not knowing who or where she is (Season 4). In other episodes, however, the series nuances such depictions of psychosis in terms of loss by foregrounding the enrichment and benefits that Carrie experiences: she is able to gather knowledge and insight during psychotic breakdowns. Initially, this is presented in the show as an enrichment at the level of subjectivity. Carrie proposes that, in a manic-psychotic state, she recovers a part of herself that is lost when her condition is treated with medication. When her father inquires about that fact that she is no longer taking her medication (in Season 2), Carrie responds, 'I wasn't myself, I was only half there'.

The fact that Carrie sees some things more clearly when she is in psychosis can nevertheless not only be attributed to the creativity and learning capacity that Lazzarato associates with the pre-individual level of subjectivity. As previously noted, the form of semiotization at this level of subjectivity also operates in the process of machinic enslavement. In this regard, it is extremely significant that Carrie refers to the skill that she gains during psychoses as 'seeing,' as this word suggests that she is not decoding or interpreting. It does not involve finding meaning or acquiring understanding, but the prediction of 'output' by examining a coherent system. During her psychoses, Carrie comprehends processes that cannot be read at the level of signifying semiotics, as the form of subjection that is operational in these processes does not decide how subjects acquire meaning in (and are subjected through) language and discourse. Instead, she 'sees' asignifying processes in which people are transformed into cogs in machinic assemblies, and in which asignifying power signs do not have meaning but produce material effects. The large paper assemblages that Carrie creates on the walls of her apartment act as diagrams. They reproduce the functioning of a machinic system, which makes it possible to predict the material effects of that system. Carrie processes large numbers of printouts into enigmatic, coloured graphs with overlaps, complicated patterns, and threads that connect different parts. These seemingly meaningless collages subsequently allow Carrie to ascertain that an attack is imminent. In sum, when *Homeland* is read with Lazzarato, the productivity of Carrie's psychoses puts forth a machinic image of the world.

### 3.3. Smoothed-Out Antagonism

The question is: what kind of machinic system does the fictional world of *Homeland* (re)present? Who or what is part of that system? Can the two forms of subjection that Lazzarato attributes to the machine (social subjection and machinic enslavement) be iden-

tified in the TV show? The way in which characters define each other in oppositional conceptual pairs (e.g., friend/enemy, mad/healthy, good/bad) indicates that a process of social subjection is occurring in the diegesis of *Homeland.* Everyone receives a role and is assigned a specific identity. Moreover, Lazzarato's interpretation of social subjection as a discursive process that is accompanied by technological and economic resources is clearly illustrated by the series. Object and subject positions for instance emerge through the use of surveillance technologies (drones, hidden cameras) and through weapons that produce victims and perpetrators. Moreover, perpetrators are detected by characters tracing money trails. Weapon production and terrorism (or counter-terrorism) are visibly embedded within an economic system.

At the same time, the roles that are assigned through social subjection in *Homeland* are continuously unsettled. Because everyone uses the same technologies and weapons, most of the subjects who perform surveillance are also the objects of spy cameras, and virtually every shooter is also struck by bullets. While Carrie repeatedly places secret cameras in other people's houses, she becomes an object of medical surveillance in Season 3, and she is spied upon in her home in Season 6. While she compassionately saves the lives of others, she also takes lives in cold blood, which makes her identity—like those of all characters in the series—changeable and ambiguous. The dissolution of boundaries between oppositions (e.g., subject/object or perpetrator/victim) could be interpreted as a process of machinic enslavement, in which everyone acts as a cog in a machinic assemblage. By using the same tactics and forms of violence, security services and terrorist organizations resemble each other to such a degree that they form a close-knit whole of elements that, in response to Lazzarato's definition of machinic enslavement, operate at the same level and cannot be articulated as binary oppositions (Lazzarato 2006, p. 4). Whereas *Homeland* surprises its viewers by repeatedly undermining binary oppositions in the first few seasons, the strength of the series after seven seasons resides precisely in the fact that the dissolution of oppositions and the slipping away of meanings is no longer surprising. What is left is the impression of an assembly of people and technologies, all involved in an ongoing flux of surveillance and interrogation, with the material 'output' consisting of moments of destruction (e.g., a bomb explosion or gas attack).

Lazzarato's theoretical-philosophical notions demonstrate that, in *Homeland*, antagonistic political groups that seem to belong to entirely different (political-cultural and/or symbolic) orders are actually all part of—and subjected to—one and the same system. Although distinct groups can be identified, all subject positions serve the operation of a single machine and are, moreover, diluted at the level of enslavement. This is not to say that there is no political dissensus between groups or that globalized power relations can arrive at a harmonious balance within a machinic system. The problem is precisely that the dissensus and inequality that characterize politics are smoothed out by machinic domination. In *Signs and Machines*, Lazzarato warns that machinic asignifying semiotics could render asymmetric power relations within heterogeneous fields formally equivalent and could forge them into a whole. In this way, power relations are depersonalized *and* depoliticized (Lazzarato 2014, p. 41), which impedes political action and emancipation.[7] In *Homeland*, plots are driven by a variety of power struggles. After eight seasons, however, these struggles appear to be empty, as no one within the networks of combatting parties wins or breaks free of the ongoing practices of surveillance and interrogation, missions and attacks, actions and counter-reactions. In an article on terrorism and trauma, Thomas Elsaesser paraphrases Jacques Derrida in proposing that:

> "A terrorists' act is a product of that which it rejects, a *mirror image* of its target. [ . . . ] The circle is almost unbreakable: terrorism and that which it is against are locked in a reciprocal game of destruction in which causes may no longer be distinguished from consequences'". (Elsaesser 2014, p. 37)

In *Homeland*, this unbreakable circle is a machine composed of many elements, which interrelate and resemble each other so profoundly that an opposition between 'terrorism' and 'that which it is against' dissolves.

Despite frantic attempts to start a new life through relocation and resignation, Carrie does not succeed in escaping the unbreakable machinic system that *Homeland* exposes throughout the course of eight seasons. Even her access to asignifying semiotics—which Lazzarato designates as both the motor behind the operation of machinic subjection and a creative source of emancipation and liberation—does not grant her any freedom. Her deliberate descent into psychosis can be understood as a conscious surrender to machinic enslavement, which allows her a clear view of the operations of the machine at a particular moment. Although this does not lead to the deconstruction of the machinic system, Carrie's manner of looking—a way of diagrammatic mapping—can be interpreted as a form of political analysis that is not primarily based on discourse analysis, ideological criticism, or a psychoanalytic approach to social-political orders.

## 4. Conclusions

By demonstrating the machinic coherence (and thus the smoothing-out and stagnation) of contemporary political antagonism, *Homeland* succeeds in forming a critical reflection on the contemporary political military-industrial arena. One question that remains unanswered after an analysis of *Homeland* based on Lazzarato, however, is how the operation of the military-industrial machine should be defined. Lazzarato's use of the word 'machine' (instead of 'ideology', 'discourse', or 'economic model') for the capitalist system emphasizes that all elements in the system have related functions that together produce something. Machines make things. In capitalist machines, social subjection and machinic enslavement are used to create a division of labour that is aimed at the production of commodities and, ultimately, profit or value-maximization. In the case of *Homeland*, it is difficult to use an over-arching term to indicate what is being produced. Although it is possible to say that a military-industrial complex is represented, the overarching principle that directs the operation of the system remains unclear. What is the logic of the machine?

The assembly of people and technologies in *Homeland* does not so much create and preserve a division of labour as it does an ongoing form of *distrust*. Every form of trust, every act of faith, is penalized or nipped in the bud. No institute, authority, or law remains intact. The representation of the contemporary 'post-9/11' military-political arena in *Homeland* can best be understood through a combination of the seemingly irreconcilable theories of Lacan and Lazzarato. With Lacan, the structural undermining of trust in Carrie's world can be regarded as a collective rejection of Names-of-Fathers. However, in *Homeland*, this does not necessarily lead to an instable social-cultural order or to the loss of subjectivity. If the series is also read with Lazzarato, distrust emerges as the structuring principle of an extensive machinic system in which various forms of subjectivity are produced and deployed, but from which no escape is possible. The goal of this mistrust is not clear, but the consequence is: destruction.

**Funding:** This research received no external funding.

**Institutional Review Board Statement:** Not available.

**Informed Consent Statement:** Not available.

**Data Availability Statement:** Not available.

**Acknowledgments:** This is a translation/reprint of "Productieve Psychoses: Psychoanalyse, Machine-analyse, en het Politieke in Homeland" originally published in Dutch by Antwerpen: Garant (Oog om Oog: Psychoanalyse en TV-Series, Houwen 2018, pp. 117–41). This translation was prepared by Janna Houwen. Permission was granted by Janna Houwen.

**Conflicts of Interest:** The author declares no conflict of interest.

## Notes

1. Because Carrie's psychotic disorder is analysed in this article as fictional, semiotic construct that produces meaning, the symptoms as well as the effects of pyschoses that are discussed in this article do not necessarily comply with contemporary psychiatric understandings of psychosis and/or bipolar disorder.

2. This article constitutes a revised and translated version of a chapter that was publisher earlier in Dutch, see (Houwen 2018).

3. The idea of 'productive pathologies' has also been discussed in relation to so called mind game films by Thomas Elsaesser (2009).

4. Lacan theorized the relation between psychosis and subjectivity approximately over the course of half a century, in which his interest (broadly speaking) shifted from identification to language and the signifier, and then to the body, desire, and the interrelations between the Real, the Imaginary, and the Symbolic. Although all of his ideas on psychosis are connected, this essay mainly draws from Lacan's lectures on psychosis in the 1950's (the third seminar), in which language and the Symbolic order hold a central position. See Vanheule's insightful *The Subject of Psychosis* (Vanheule 2011) for an extensive discussion of different phases in Lacan's oeuvre.

5. Carrie's familial and educational background have recently received literary treatment in Kaplan's official "Homeland" novels. The details that are added to the Carrie's biography through this process of transmedial storytelling remain outside of the scopeof this article. For more information on the novelizations, see (Sobreira 2021).

6. Other ways to gain access to the order of asignifying semiotics are, according to Lazzarato, drug use, being in love, and going through an existential crisis (Lazzarato 2014, p. 104).

7. Lazzarato examines possible modes of political subjectivity and emancipation that allow for an escape from the enslaving clutches of the capitalist system in the final chapter of *Signs and Machines* (Lazzarato 2014).

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
