# Peer review of "Productive Psychoses: Views on Terrorism and Politics in Homeland"

_humanities, doi:10.3390/h12030037_

Round 1

Reviewer 1 Report

The article presents a relevant study. However, seven points require some attention:

I. The manuscript needs spell check because it contains a few typographical errors, as well as punctuation and grammar problems, namely in the following passages: "She keeps evades the rules" (line 184); "especially confirmed in when, in Season 5" (188); "brings herself or others in danger" (197); "She can turn into an understanding friend in order to gather information from the wife a Hezbollah leader, or plays the part of a comforting lover" (226-227); "frantic attempts to use communicate experiences and ideas" (306); "but their ideological patriarchal structures are irendered instable by the moral doubts of members" (338-9); "as so called ‘power signs’ they work directly on material components" (427); "on the one handf" (481); "this is presented in the show an enrichment at the level of subjectivity" (486); "Al-Airan" (608).

II. The article would benefit from a sort of “disclaimer” in the introduction stating clearly that the traits and symptoms of psychosis and / or bipolar disorder discussed in the text merely have the nature of analytical hypotheses since they are not actual conditions observed empirically. It should be noted that both the investigated subject (Carrie Mathison) and the context (the whole “Homeland” television universe) in which she is inserted are merely fictional constructions with only discursive links with the empirical dimension. In this way, the diagnoses and conclusions extracted from this fictional work are merely conjectural given their linguistic and rhetorical nature and, for this reason, they do not necessarily reflect empirical traces of the conditions described. If "the [actual human] subject is an effect of language", a fictional character in a TV series has an even more linguistically precarious and imaginary ontological status.

III - Since the series has already concluded and it is no longer being shown on television, the study needs to present more details about this particular Showtime production. The reader of future generations may have difficulties in understanding the object being studied if more details are not made available in the text (director, creator, cast, original release, channel, among others). Such information can easily be included in the research by means of a simple footnote. A recent study gathers a lot of information about the series: "The Adaptation of Homeland from Screen to Page: Challenges of Two Novelizations Based on the Television Series” (https://proa.ua.pt/index.php/rual/article/view/28162/20227). Also, whenever the manuscript mentions a key event in the television series, it is important to cite the season and the episode numbers (e. g. in the fourth episode of season 6…). Such clear indications are very useful to thoughtful readers.

IV. The manuscript lacks more recent scholarship, particularly articles and books published this decade. In terms of the references, they require some revision since they do not seem to follow the general formatting guidelines. Also, the reference “Horsman, Yasco (2018)” does not seem to formally appear in the body of the text.

V. The analysis of “Homeland” seems to rely too heavily on political statements and on dated studies associated with postcolonialism. Although the arguments aim at problematizing dichotomies such as “healthy / sick, civilized / primitive, rational / emotional, adult / child, White / Black, Christian / Muslim, good / bad, victim / perpetrator”, the proposed analysis tends sometimes to incorporate these same models not only as valid but also as productive categories of interpretation. It seems to be the case when the text emphasizes the "tensions and conflicts between Western and non-Western groups" (lines 41-42). If the point is that dichotomies are the whole source of conflict and they tend to lead to “meaninglessness”, it seems incoherent to classify the factions in “Homeland” within these two same vast “stereotypical” blocs. The overall discussion would benefit from abandoning the postcolonial angle altogether and focusing more on the specificities of the series. The argument that Carrie identifies a terrorist in Brody because she is “the only one who manages to see [him] […] outside of the customary framework of a White, male American war hero” does not seem to take into consideration the enormous amount of intelligence that she collected and examined until she got to that conclusion. It is not a mere question of avoiding social and racial profiling when dealing with suspects as the argument implies.

VI. The manuscript seems to suggest that Carrie’s condition is triggered by “crossing the border between Western and non-Western cultures” because it “is often presented as ‘dangerous business’ in Western tales of colonized areas" (lines 44-6). Carrie Mathison is a very complex character whose biography is not confined only to the television series. Through a process of transmedia storytelling, the details of her life are also developed in Andrew Kaplan’s official “Homeland” novels. There is a whole emotional, educational and familial background to Carrie that received a literary treatment recently. A summary of those novels can be found in recent studies such as the paper “The Adaptation of Homeland from Screen to Page: Challenges of Two Novelizations Based on the Television Series” [journal: RUA‑L, n. 10, s. II, 2021, pp. 107‑130, ISSN 0870‑1547] (Available at: https://proa.ua.pt/index.php/rual/article/view/28162/20227). The books were produced in co-authorship with the “Homeland” showrunners and what these novels show is that Carrie experienced those “productive psychoses” since her early adulthood. These episodes do not necessarily have to do with her “crossing borders” or having to come to terms with a sense of “otherness” in Eastern cultures. That is why the postcolonial argument is hard to buy. It somehow oversimplifies one of the most complex characters in television. The manuscript hints at this difficulty when it states that “it is impossible to attribute Carrie’s psychotic delusions unequivocally to the madness of the cultural others that she meets” (lines 88-9). This idea should be further emphasized in the article.

VII. The use of two “completely different theoretical perspectives” in the study seems confusing. Although the author makes an effort to render these two approaches compatible, there seems to be an inherent strangeness about this combination. Sub-textually, the manuscript itself suggests that Lazzarato is subpar compared to Lacan: “Lazzarato has not written any detailed studies on psychosis” (line 448); “Lazzarato does not explain why psychosis exactly provides access to asignifying semiotics” (456); “THERE APPEARS TO BE a general correspondence between the two theoreticians when it comes to psychosis” (470). As they are very “different”, the positions assumed in the text are ambivalent since the author not only implies but also reinforces the notion that they sustain antagonistic views. The Lazzarato portion of the text does not help much to analyze “Homeland”. As the article suggests, Lazzarato’s theory seems to follow the formulaic “capitalism is machine” metaphor derived from the silent era of communist cinema (Chaplin’s “Modern Times”, Fritz Lang's “Metropolis”, to name a few). The Italian philosopher seems to ignore that this “productive machine” is not exclusive of capitalist societies (cf. von Donnersmarck's “The Lives of Others” or Agnieszka Holland's “Mr. Jones”). Thus, the manuscript would be less conflicting if it concentrated solely on the Lacanian aspects of the analysis and suppressed the entire Lazzarato section. If the author chooses to maintain both, however, the text requires revision to “hide” some of these discursive marks of ambivalence from the readers.

Author Response

I. I have removed the spelling and grammar errors as well as double spaces. 

II. I added the proposed disclaimer in a footnote (note 1)

III. I included the suggested source in a footnote (note 3) and also added it to the list of references

IV. The source mentioned here (Horsman) did not appear in the body of the text indeed, I have removed it from the list of references. I have not added any more recent scholarship as I feel the text includes quite a number of recent publications (e.g., Al-Airan 2012, Bevan 2015, Letort 2017, Zanger 2015)

V. The reviewer mentions here that the article would benefit from abandoning the postcolonial angle altogether. I agree, this is precisely what I aim to do, in line 44-115 I explain why and how. 

VI. Added to footnote 3

VII. I reread my remarks and I am not sure how to resolve the "inherent strangeness" of the combination of Lacan and Lazzarato that reviewer one points out. For me, this strangeness is what makes the article interesting: two very different theoretical models can still be made to 'work together' because in spite of great disparities they have one thing in common: they deal with meaninglessness (yet in different ways). 

Reviewer 2 Report

I have carefully read your paper twice and I am not sure if I thoroughly understood Lacan's theory. You have chosen a very interesting subject: Carrie Mathison's mental issues that allow her to become a very efficient CIA operator. Whether one likes Lacan's philosophy or not, is not the issue here, as you have clearly stated that  Lazzarato's thought is a better tool to analyse her psychosis. I thoroughly enjoyed reading your paper, but I am not sure if I properly understood why Lacan's psychoanalysis is somewhat inefficient or a weak too to analyse her psychosis. You focus on Mathison's psychosis which the series Homeland presents as a major characteristic feature of her's. Yet, in the last series, when she defects to Moscow with Yevgeni Gromov to replace Anna and become Saul's new asset at the GRU HQ, she does not display any psychosis, much as if she was healed. Could you add some thoughts about the last series 8 and the reasons why Carrie is acting like a normal, healthy person? I think that such a brief paragraph would be beneficial for your paper - and the reader who knows the series. Furthermore, I did not quite understand Lacan, and I think that your paper would benefit from additional explanations of both Lacan and Lazzarato's perspectives. But overall, it's excellent scholarship and the main research interest is a fresh and new approach to Homeland and Carrie's psychosis.

Author Response

Thank you for this feedback. Lacan's theoretical ideas are very complicated indeed, and I can understand the reviewer's struggles in understanding Lacan. I have tried to add a few more explanatory remarks (in section 2.1 and 2.4). However, I did not add very extensive additional explanations as this would only make the article more dense to read, I believe.